# A Newly Discovered Forest of the Whip Coral *Viminella flagellum* (Anthozoa, Alcyonacea) in the Mediterranean Sea: A Non-Invasive Method to Assess Its Population Structure

**DOI:** 10.3390/biology11010039

**Published:** 2021-12-28

**Authors:** Giovanni Chimienti, Ricardo Aguilar, Michela Maiorca, Francesco Mastrototaro

**Affiliations:** 1Department of Biology, University of Bari Aldo Moro, 70125 Bari, Italy; m.maiorca@studenti.uniba.it (M.M.); francesco.mastrototaro@uniba.it (F.M.); 2CoNISMa, 00196 Rome, Italy; 3Oceana, 28013 Madrid, Spain; raguilar@oceana.org

**Keywords:** Anthozoa, animal forest, gorgonian, biometry, mesophotic, ROV, image analysis, IMOD, habitat, Aeolian Islands

## Abstract

**Simple Summary:**

Some corals belonging to the orders Alcyonacea and Antipatharia have elongated, unbranched shapes, and are generally addressed as sea whips. The octocorals *Viminella flagellum* are the main sea whip species inhabiting the Mediterranean Sea, where they can form large colony aggregations known as coral forests. These habitats are of great conservation importance; they provide a suite of ecosystem goods and services, and their monitoring is essential to plan appropriate conservation strategies. One of the most important indicators on the status of a coral forest is its population structure, such as the frequency of different size classes within the coral population. This is very difficult to assess in sea whips because of the length and high flexibility. Here, we report on the findings of a newly discovered, monospecific forest of *V. flagellum* in Aeolian Archipelago (Mediterranean Sea), and we present a new method to study its population structure using video analysis. The results of the survey indicate that the Aeolian coral population was in good condition, without significant anthropogenic impacts. The new method presented here proved to be an effective and promising tool for the monitoring of this vulnerable marine ecosystem. It can be applied to every known population of *V. flagellum* as well as adapted to other sea-whip species.

**Abstract:**

Coral forests are vulnerable marine ecosystems formed by arborescent corals (e.g., Anthozoa of the orders Alcyonacea and Antipatharia). The population structure of the habitat-forming corals can inform on the status of the habitat, representing an essential aspect to monitor. Most Mediterranean corals live in the mesophotic and aphotic zones, and their population structures can be assessed by analyzing images collected by underwater vehicles. This is still not possible in whip-like corals, whose colony lengths and flexibilities impede the taking of direct length measurements from images. This study reports on the occurrence of a monospecific forest, of the whip coral *Viminella flagellum* in the Aeolian Archipelago (Southern Tyrrhenian Sea; 149 m depth), and the assessment of its population structure through an ad-hoc, non-invasive method to estimate a colony height based on its width. The forest of *V. flagellum* showed a mean density of 19.4 ± 0.2 colonies m^−2^ (up to 44.8 colonies m^−2^) and no signs of anthropogenic impacts. The population was dominated by young colonies, with the presence of large adults and active recruitment. The new model proved to be effective for non-invasive monitoring of this near threatened species, representing a needed step towards appropriate conservation actions.

## 1. Introduction

The whip coral *Viminella flagellum* (Johnson, 1863), is an Atlantic-Mediterranean species living in temperate waters [1]. The taxonomic position of this species is currently considered uncertain and possibly attributed to the genus *Ellisella*, although it is still to be clarified, the binomial *V. flagellum* will be used in this paper. It lives under dim-light conditions, in the mesophotic zone, or in the total darkness of the aphotic zone [2]. This species belongs to the family Ellisellidae, and it is characterized by unbranched, monopodial colonies that occasionally show one or a few bifurcations, and can reach up to 3 m in height in the Atlantic Ocean [1,2,3,4]. Polyps are small, cylindrical, and retractile into calyces. As with many other octocorals, *V. flagellum* can form dense, monospecific, or mixed aggregations, broadly known as coral forests [5,6,7,8,9]. This habitat is often present in high-energy environments, such as seamounts (especially on their plateau), because sea whips are generally adapted to turbulent conditions thanks to their shape, their high flexibility, and their thick basal stem [10].

*V. flagellum* is mostly found along the Eastern Atlantic (Azores Islands, Cape Verde Islands, Canary Islands, Madeira, Josephine Bank, Great Meteor Bank and Moroccan coast), particularly below 350 m and down to approximately 1000 m depth [1,3]. In the Mediterranean Sea, these octocorals have been reported mainly in the western basin, on mesophotic rocks surrounded by detritic bottoms at 90–200 m depth and, occasionally, down to 500 m depth [9,11]. Forests of *V. flagellum* have been patchily observed on seamounts in the Alboran Sea and around the Balearic Islands [11,12], along the French canyons and Corsican coasts [1,13], southern Sardinia [5,6,14], and northwest Sicily [15].

Coral forests are considered vulnerable marine ecosystems (VMEs) because of their lives–history traits, including slow growth and high longevity, as well as their slow (to zero) recovery capacity after anthropogenic disturbances [16]. As with other forest-forming corals, *V. flagellum* colonies can be affected by fisheries, although quantitative data about incidental catches of this species by bottom contact fishing gear are still scarce [17]. However, some dead colonies, due to entanglement in lost fishing lines, have been recently observed along the Sardinia coast [18]. *V. flagellum* is also listed as near-threatened on the Red List by the International Union for the Conservation of Nature (IUCN) [19], deserving proper monitoring and conservation actions. Although several actions have been recently taken towards monitoring the sensitive habitats of the Mediterranean Sea through, for instance, the EU Marine Strategy Framework Directive [20], international initiatives do not include coral forests yet.

Together with distribution and abundance data (e.g., [5,7,14,21,22,23]), assessing the population structure of habitat-forming corals can provide valuable information about the conservation status of the habitat and its associated community, e.g., [24,25,26]. However, it is often difficult to identify methods aiming to study coral habitats using non-invasive techniques, such as scuba diving, towed/drift/drop cameras, and remotely operated vehicles (ROVs). Arborescent corals, for example sea fans and black corals, can be measured considering height and width [25,26,27,28], while length and number of polyp leaves can represent a valuable proxy of size and biomass in sea pens [24,29]. Although monopodial, sea whips are typically long and very flexible, and they often bend backward and forward, occupying different spatial planes with respect to the observer. For this reason, size assessment based on images is biased by the perspective in the photogram, while size references (e.g., laser beams) are present only on one spatial plane. This study reports the finding of a *V. flagellum* forest observed off Stromboli Island (Aeolian Archipelago, Mediterranean Sea). Moreover, the study aims to assess the population structure of the species through the development of a new model to measure the length of the colonies using non-invasive video imaging. Based on the assumption that colony width does not vary considerably in this species, as for other octocorals with similar morphology [30], we used the mean width as size reference of the different portions of each colony, regardless of their position in space, in order to estimate the length of the colonies from ROV videos. The model proved to be an effective tool for the monitoring of *V. flagellum* populations.

## 2. Materials and Methods

From May to June 2018, the seabed around the Aeolian Islands (Italy, southern Tyrrhenian Sea) was investigated aboard the Oceana’s Ketch Catamaran *Ranger*. One site, 1.1 km off Stromboli village (38°48.3797′ N–015°15.3256′ E), revealed a monospecific population of *V. flagellum* that was considered in this study. The seafloor surrounding Stromboli Island is characterized by volcanic outcrops, seamounts, gullies, submarine channels, and canyons down to more than 2000 m depth, with the presence of hydrothermal activity from shallow to deep waters linked to the active Stromboli volcano [31].

The survey was carried out using a Saab Seaeye Falcon DR ROV equipped with a high-definition video (HDV) camera of 1920 × 1080 resolution, 1/2.9″ Exmor R CMOS Sensor, minimum scene illumination of 3–11 Lux, and a 4.48 mm, f/1.8–3.4 zoom lens [32,33,34]. The ROV also hosted a depth sensor, a sonar, a compass for underwater navigation, as well as two laser beams providing a 10-cm scale for size measurements. The ROV position was recorded every second using a LinkQuest TrackLink USBL Transponder with up to 0.25° accuracy.

The portion of the seabed characterized by the presence of a forest of *V. flagellum* was considered in the video analysis (ca. 2 min of video recording). Videos were processed using ImageJ software by defining sampling units of 2.5 ± 0.2 m^2^, according to the minimal area used for visual surveys on mesophotic bottoms [25,34,35,36,37]. Frames with bad visibility, or where the ROV camera was not oriented properly for a correct estimation of the area based on laser pointers, were discarded. The population of *V. flagellum* was quantified both by occupancy (frequency of occurrence in the set of sampling units) and by abundance (number of colonies per sampling unit), then the density (colonies m^−2^) was calculated for each sampling unit and expressed as mean ± standard error. Epibiosis and number of branching colonies were also quantified.

High-resolution still images for morphometric analysis were extracted directly from the ROV footage in order to measure the height of the colonies whose position and ROV framing allowed it. Each frame was enhanced using ImageJ/Fiji software to reduce the image defects (chromatic aberration, interlacing effects). In particular, interlacing was the most disruptive effect in the images, as it is unnoticeable when watching a video but relevant when a single frame is extracted from a video, impairing accurate measurements (see [38] for a rationale). For this reason, the extracted frames were de-interlaced [39] by vertical interpolation using the Fiji plugin for ImageJ. Images were then analyzed using IMOD software [40].

Colonies of different sizes that showed an optimal position compared to the ROV camera and laser pointers (i.e., presence on the same plane of the lasers and standing position of the colony without facing forward or backward) were selected (Figure 1a). Dead leaves of the seagrass *Posidonia oceanica*, randomly present on the bottom, were also used as size reference when present in close proximity of specific colonies and oriented in the correct plane. *P. oceanica* leaves have an almost-constant width of 1 cm which may slightly vary by about 1 mm according to the season and the geographic area [41]. This value was also checked and validated in those frames where laser pointers were present. Based on the scale provided by laser pointers and/or by size references on the bottom, length and width (cm) of all the measurable colonies (n = 21) were measured using IMOD (Figure 1a–c). In particular, width measures were taken at the proximal, the median and distal portion of each colony, then the mean width value (µ_w_) was calculated. The Full Width at Half Maximum (FWHM) of Gaussian shaped grayscale density was considered in order to increase the accuracy of width measurements and exclude those pixels affected by partial volume effect and defocus aberration (Figure 1d–f). Eventual errors in size measurement are in the order of millimeters and have been considered negligible with respect to the 20-cm size classes.

Colonies of *V. flagellum* are characterized by monomorphic and highly contractile polyps, so width measures were collected in-vivo considering only the base of the calyces, regardless of the contraction/relax status of each colony. Assuming that colony widths do not vary significantly with respect to the height, µw was used as fix parameter in the population in order to estimate the height of the colonies clearly visible from the ROV images, also when they were not present entirely in the same plane of the size references (e.g., colonies positioned backward or forward).

Width measurements in sub-pixels (w_i_ (px)) were taken in at least three portions (proximal, median, and distal) for every *V. flagellum* colony that was visible in its entire length, regardless of its position in space. The measurement was carried out using IMOD and following the abovementioned procedure. The width of each colony was assumed to correspond to the mean width μ_w_ (cm) calculated for the population. The total length of each colony was then estimated as follows:h_i_ (cm) = µ_w_ (cm) × h_i_ (px)/w_i_ (px),(1)
where h_i_ (cm) and h_i_ (px) express the length of the same i-th colony in cm and in sub-pixel, respectively, while w_i_ (px) is the mean width of the i-th colony measured in sub-pixels. Eight size classes of 20 cm each were identified, from 0.1–20.0 to 160.1–180.0 cm. The estimated length values were used for the assessment of the population structure of *V. flagellum* in the study area, analyzed in terms of skewness and kurtosis calculated by means of the R software functions *agostino.test* [42] and *anscombe.test* [43].

## 3. Results

### 3.1. A Forest of Viminella flagellum

A forest of *V. flagellum* was found on a rocky outcrop at 149 m depth (Figure 2a), and a total of 777 colonies were counted over an area of 40 m^2^ (16 sampling units analyzed). Colonies showed 100% occupancy, with each sampling unit characterized by at least one colony. A mean abundance of 48.6 ± 0.4 colonies per sampling unit (mean ± standard error) was observed, with a density of 19.4 ± 0.2 colonies m^−2^ (maximum density of 44.8 colonies m^−2^).

About 6% of the colonies showed at least one ramification, more often with a single bifurcation, in some cases with two branches (four gorgonians), only in two cases with three branches. Furthermore, one of the latter showed an unusual characteristic with three branches originating in the same point (Figure 2c–e).

The presence of epibiont characterized 26% of the colonies, with the dominant presence of serpulids belonging to *Filograna/Salmacina* complex (Figure 2b). The highest values of colonies with epibiosis were recorded in those sampling units also characterized by high densities of *V. flagellum*, suggesting that the tissue abrasion due to the direct contact between different colonies could facilitate the settlement of epibionts. When some colonies were particularly close to each other, their distal portions were often entangled, so it was not possible to measure their size and distinguish their terminal parts.

### 3.2. A New Method to Assess the Population Structure

Only 21 colonies were properly positioned with respect to the ROV camera and the size references, resulting measurable with standard methods (Figure 2a,b,d). A mean width value (µw) of 0.35 ± 0.01 cm was calculated based on these colonies. Then, by applying (1) it was possible to estimate the length of 232 colonies based on their width (Figure 3c,e).

The size-frequency distribution showed that 58% of the colonies were smaller than 40 cm (Figure 4), representing mostly juvenile colonies. The distribution was platykurtic and highly skewed, with an evident right tail represented by large adults up to 170.5 cm in height. To our best knowledge, this latter represents the highest value of colony height reported for *V. flagellum* in the Mediterranean Sea.

## 4. Discussion

Forests of *V. flagellum* have been sporadically observed in the Mediterranean Sea, so that their general distribution is not yet comprehensively known [5,6,7,9,14]. This lack of information is related to the lack of exploration in many submarine canyons of the basin as well as the patchy distribution of *V. flagellum*. This study reports the occurrence of a population of these mesophotic octocorals at Stromboli Island, with remarkable density over a relatively small area. Considering the patchy distribution of *V. flagellum*, which tends to occupy the top of rocky outcrops on detritic bottoms, it is possible that more subpopulations are present in the nearby area of the Aeolian Archipelago, although we observed this species only in one specific ROV transect. The high density is not surprising for this species, which is known to reach up to 60 colonies m^−2^ in the basin [5,7], although very dense populations have been rarely observed thus far. Such density can also be the result of asexual reproduction, since in many sea whips the distal portion of the colony may be coiled, forming a spiral structure that may detach generating young colonies [44,45]. High densities cause the entanglement of the different colonies and it seems to enhance epibiosis because of mechanical abrasion of the tissues. Although the incidence of epibiosis is generally higher in stressed coral populations, in this case, the high density suggests the presence of stable environmental conditions and the absence of significant impacts. In fact, no lost fishing gears or other anthropogenic impacts were observed within the *V. flagellum* forest.

The reciprocal entanglement of long colonies in dense populations can represent a form of intraspecific competition. However, this phenomenon might limit the length measurement of the colonies involved and could bias a size-frequency distribution by excluding some of the longest colonies.

In order to solve this bias, we considered for the biometry only those sampling units where most of the colonies were potentially measurable using IMOD, in order to prevent the selection of small colonies with respect to the largest. For this reason, the observed dominance of young colonies reflects, in our opinion, the real condition of the population. Such highly skewed size-frequency structure can be linked to sexual reproduction and active recruitment, as occasionally observed in other gorgonian species such as *Eunicella singularis* [26], as well as in reef-forming corals [46,47]. Despite healthy gorgonian populations tending to show a peak at the intermediate classes (e.g., [26,37]), differences observed between gorgonian species may be explained by different reproductive strategies. In our case, *V. flagellum* is a broadcast spawner with continuous gamete presence and overlapping reproductive cycles [48] that can positively influence local recruitment.

Although Mediterranean colonies of *V. flagellum* are reported to be usually less than 1 m high and characterized by a thinner basal diameter of the stem compared to Atlantic ones [1,3], the presence of large, adult colonies up to more than 1.70 in height suggests that information about the biology of the species is still incomplete. In addition, branch dichotomization is not as uncommon as previously thought [1] and, sometimes, *V. flagellum* can show trifurcation of the branches.

The lack of validation based on physical specimens or other replicates from different populations represents a recognized limit in the method. However, *V. flagellum* occurs at depths not accessible by scuba diving, thus in-vivo direct measurements on Mediterranean colonies are currently not possible. Moreover, its populations are difficult to observe and sensitive to destructive sampling practices, so the collection of samples for validation purposes would affect the studied population. Our mean width value (µw) of 0.35 cm is in accordance with the one reported for colonies from the Sicily Strait (approximately 2–3 mm, based on 1 sample and 10 observed colonies) [49]. To the best of our knowledge, there are no other width estimates carried out on Mediterranean colonies. Further perspectives include the improvement of the method based on other populations of *V. flagellum* from both Mediterranean Sea and Atlantic Ocean, as carried out for other Atlanto-Mediterranean species [28], as well as its application to monitor coral forests and assess the effects of anthropogenic impacts, such as fisheries and global warming [22,23,50]. The method presented here can be also adapted to other whip-like corals, such as gorgonians and black corals at tropical latitudes [4].

## 5. Conclusions

The whip-coral forest found in the Aeolian Archipelago is one of the very few monospecific forests of *V. flagellum* known thus far. Its finding represents a further step to the still-incomplete knowledge that understates the geographic and bathymetric distribution of the species. Analyses of corals’ size-frequency distribution can reveal important characteristics on these VMEs. Our new method to assess the size of *V. flagellum* colonies represents a further step to achieve a non-invasive study and monitoring of essential coral populations in mesophotic and deep areas, in order to improve our understanding of these vulnerable habitats. Moreover, in the frame of current international initiatives targeting the monitoring of sensitive and important habitats and their conservation (e.g., EU Marine Strategy Framework Directive, EU Habitat Directive), these tools can inform and support appropriate actions for strategic environmental protection and a sustainable management of human activities.

## Figures and Tables

**Figure 1 biology-11-00039-f001:**
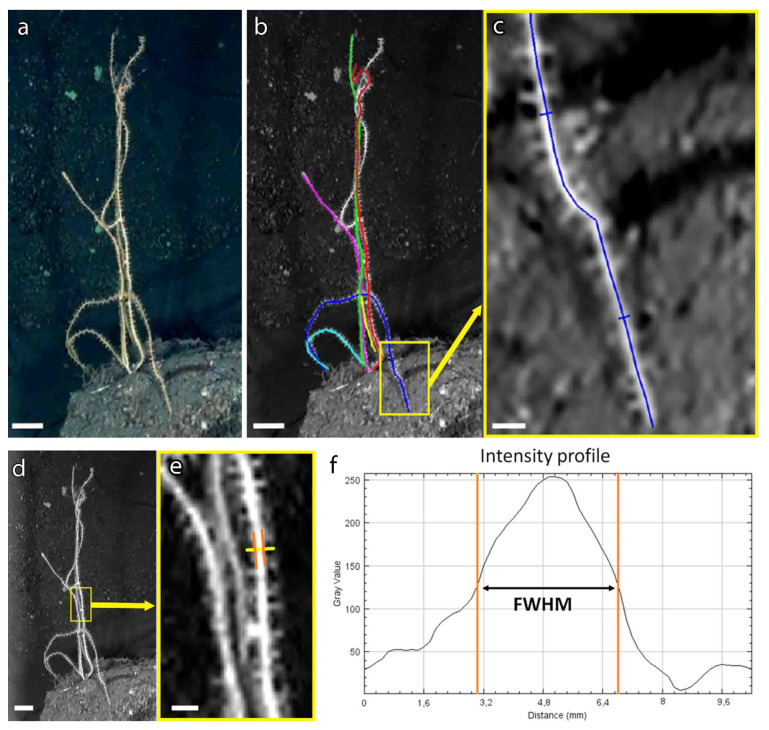
Biometric measurement of *Viminella flagellum* from ROV video. (**a**) Seabed with several colonies; (**b**) identification of each colony and length measure for those clearly visible (colored segments follow the entire length of each colony); (**c**) detail of the proximal portion of a colony and the measure of its width; (**d**) middle portion of the same group of colonies with (**e**) detail of the measurement of the middle width of a colony; (**f**) full width at half maximum (FWHM) graph. Scale bars: (**a**,**b**,**d**): 5 cm; (**c**,**e**): 1 cm.

**Figure 2 biology-11-00039-f002:**
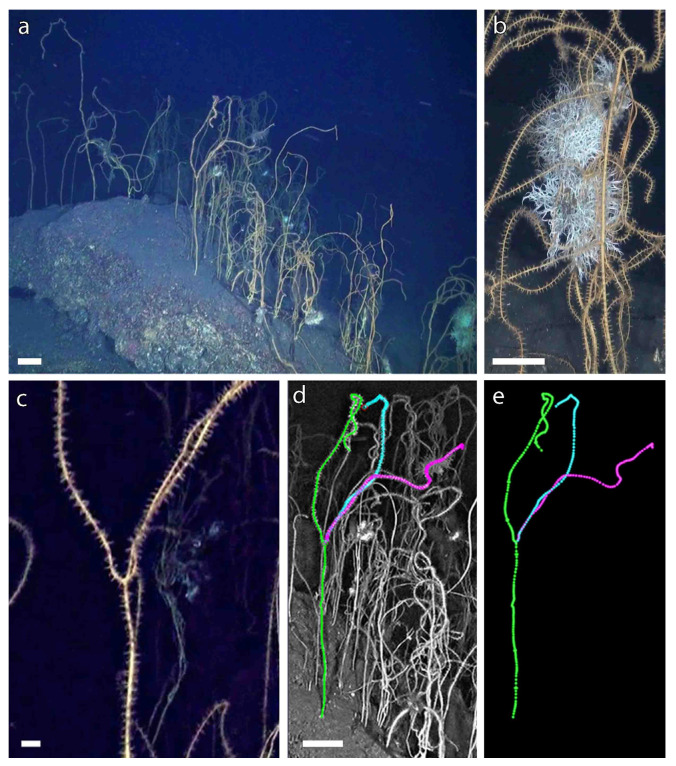
Forest of *Viminella flagellum*. (**a**) Overview of the habitat; (**b**) detail of entangled colonies with epibionts of the *Filograna/Salmacina* complex; (**c**) Trifurcated colony with (**d**) its identification within the forest and (**e**) isolation for the biometric measure. Scale bars: (**a**,**b**,**d**): 10 cm; (**c**): 1 cm.

**Figure 3 biology-11-00039-f003:**
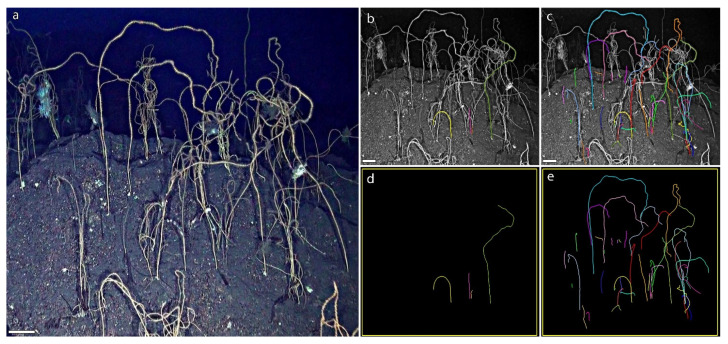
Example of application. (**a**) Frame from the forest of *Viminella flagellum*. Staining of colonies (**b**) measurable with usual ROV imaging and (**c**) measurable with the method here presented; (**d**,**e**) Isolating and highlighting the respective measurable colonies with the two methods. Scale bars: 10 cm.

**Figure 4 biology-11-00039-f004:**
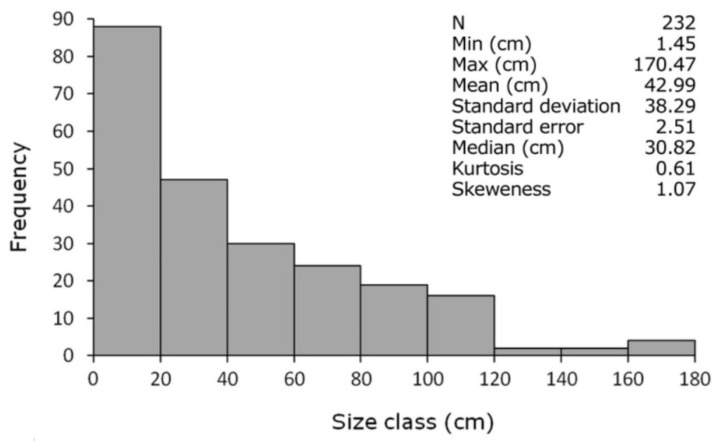
Size-frequency distribution of *Viminella flagellum* at Stromboli Island.

## Data Availability

The dataset generated during and/or analyzed for the current study is available from the corresponding author upon request.

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
