# Peer review of "A Newly Discovered Forest of the Whip Coral Viminella flagellum (Anthozoa, Alcyonacea) in the Mediterranean Sea: A Non-Invasive Method to Assess Its Population Structure"

_biology, 2021, doi:10.3390/biology11010039_

Round 1

Reviewer 1 Report

This paper presents an innovative approach to assess the population structure of a deep gorgonian species and to assess the conservation state of these populations.  The authors propose an original method to analyse the ROV images. This method provides new data about the density and the demographic structure of this species. 

Nevertheless, two remarks about the methodological aspects, results and discussion:

1- To define a scale you directly use laser or the width of dead leaves of Posidonia oceanica. Indeed, its values are more or less 1cm but precise measures show that this parameter can vary of 20 to 30% according to the season and the location of Posidonia meadow (Gobert, 2002; Gobert et al. 2009). In this context,  peharps this variation can bring a bias in the final measurement of colonies. 

2- the number of colonies selected to precise the µw value seems to be low (21 colonies). Another data could be necessary to precise this value....(Figure a,b,d) need to be change as (Figure 2 a,b,d).

3- In the discussion (line 213), you precise that monospecific forests of Viminella flagellum have been rarely observed all over the Mediterranean sea but in the introduction, you mentioned that these populations have noted in Alboran, the balearic islands, the French coasts (Viminella forests are well distributed along the western part of Corsica), in Sardinia and Sicily. This lack of data (and knowledge) is probably due to (i) the lack of exploration in the submarine canyons in the other parts of Mediterranean sea and (ii) the type of distribution of Viminella flagellum (located patches).

In conclusion, this paper can be published after minor corrections 

Author Response

We are grateful to the reviewer for the comments and suggestions provided. Here a point-by-pont reply to the remarks about the methodological aspects, results and discussion:

1- To define a scale you directly use laser or the width of dead leaves of Posidonia oceanica. Indeed, its values are more or less 1cm but precise measures show that this parameter can vary of 20 to 30% according to the season and the location of Posidonia meadow (Gobert, 2002; Gobert et al. 2009). In this context, perhaps this variation can bring a bias in the final measurement of colonies.

Reply: We thank you the reviewer for the comments and suggestions. We implemented this part of the text to consider also possible variations in the width of P. oceanica leaves that, however, is in the order of decimal centimetres, being negligible in our case where we have size classes of 20 cm. We added: “Dead leaves of the seagrass Posidonia oceanica, randomly present on the bottom, were also used as size reference when present in close proximity of specific colonies and oriented in the correct plane. Despite the width of P. oceanica leaves can slightly vary of 1 mm according to the season and the geographic area (authors personal observation), leaves were considered having an almost-constant width of 1 cm [41]. This value was also checked and validated in those frames where laser pointers were present.”. We did not find any specific information related to such variation in the references suggested. In Gobert et al. 2002 it is reported a certain variability in leaves surface area but it seems to be related to leaves length rather than leaves width. However, since this information is not clear we do not feel confident in relating our study to it.

2- the number of colonies selected to precise the µw value seems to be low (21 colonies). Another data could be necessary to precise this value....(Figure a,b,d) need to be change as (Figure 2 a,b,d).

Reply: We recognize that, despite the abundance of colonies observed, only 21 were in the right position to calibrate the model. This limit has been highlighted in the discussion and future perspectives include certainly a better calibration of the model with other colonies. Meanwhile, this study sets a baseline since, with the exception of few colonies measured by Carpine & Grasshoff 1975, V. flagellum colonies have never been measured before, because of the objective difficulty of carrying direct or indirect measurements. We added the following text to the discussion: “The lack of validation based on physical specimens or other replicates form different populations represent a recognized limit in the method. However, V. flagellum occurs at depth not accessible by scuba diving, thus in-vivo direct measurements on Mediterranean colonies are currently not possible. Moreover, its populations are difficult to observe and sensitive to destructive sampling practices, so the collection of samples for validation purposes would affect the studied population. Our mean width value (µw) of 0.35 mm is in accordance with the one reported for colonies from the Sicily Strait (approximately 2–3 mm, based on 1 sample and 10 observed colonies) [49]. To the best of our knowledge, there are no other width estimates carried out on Mediterranean colonies. Further perspectives include the improving of the method based on other populations of V. flagellum from both Mediterranean Sea and Atlantic Ocean, as carried out for other Atlanto-Mediterranean species [28], as well as its application to monitor coral forests and assess the effects of anthropogenic impacts, such as fisheries and global warming [22,23,50].”

Figure changes done.

3- In the discussion (line 213), you precise that monospecific forests of Viminella flagellum have been rarely observed all over the Mediterranean sea but in the introduction, you mentioned that these populations have noted in Alboran, the balearic islands, the French coasts (Viminella forests are well distributed along the western part of Corsica), in Sardinia and Sicily. This lack of data (and knowledge) is probably due to (i) the lack of exploration in the submarine canyons in the other parts of Mediterranean sea and (ii) the type of distribution of Viminella flagellum (located patches).

In conclusion, this paper can be published after minor corrections 

Reply: We implemented the text according to the suggestions of the reviewer.

Reviewer 2 Report

Review: Chimienti et al. A new forest of the whip coral Viminella flagellum (Anthozoa, Alcyonacea) in the Mediterranean Sea: a non-invasive method to assess its population structure

Overall

The manuscript describes the identification and analysis of a whip coral population off the island of Stromboli in the Mediterranean Sea. The data was collected by ROV video. The analysis included estimation of population density and structure. The analysis of structure is made from image analysis, by creation of a relationship for length-width on part of the population then applying the relationship through width measurements to create a size frequency analysis for the whole sampled population. The manuscript is relevant to the journal, but I have some issues with validity of how the method is applied, mostly related to lacking a solid validation. The point of the methodology is that it is non-invasive so the sensitive species is not harmed during measurement, but in the overall analysis the method uses measurements created from a subset of the images/population to apply to the whole but limited population observed. There is no mention of validation, on any physical specimens or another population – this point needs to be absolutely addressed.

There are two further points also that need to be addressed because you are presenting a measurement methodology: accuracy and resolution - and this runs into error and error propagation:

  1. How sure are you that the laser spots were always in the correct plane and perpendicular to the camera nest to the coral for 21 colonies. (it would be useful in at least one of the photos to show the laser spots highlighted if not easily seen.
  2. The measurements are primarily based/calibrated on two parallel laser spots set at 10 cm distance. In section 3.1 calculated measurements go to 5 decimal places. However no calibration accuracy measurement is given (just standard error on the derived measurements) What is the potential error in a range of consecutive images for the laser spots.
  3. Error uncertainty propagated by the different issues. Errors behind calibrating the scale from the lasers or against Posidonia leaves, errors with the measurements with on-screen software,

I do accept that the errors are reduced or not important when the measurements are binned into the different and wide size classes – but this needs to be stated.

Your method the major point to the paper is hardly discussed and the targeted results – density and size distribution are not compared in detail to other populations, leaving the reader to have to look up the references to find this information.

Without addressing these points, I would not recommend the manuscript for publication.

Other edits/comments:

Simple Summary

Line 20: change to : “… Results of the survey indicated that the Aeolian coral population was in good condition and…”

Abstract

Line 26: syntax problem, the population structure/status of the coral does not result in monitoring actions.

Line 26: change to “Most Mediterranean corals….”

Line 29: change to “…flexibility impedes the taking of direct …”

Line 30: change to “…Viminella flagellum in the Aeolian…”

Line 34: change to “…the presence of large adults…”

Introduction:

Line 43: WORMS gives your given species name as “uncertain” but the accepted name is Ellisella flagellum (Johnson, 1863). I do not think that “possibly synonimised” is a correct way of referring to the species. Can you please clarify.

Line 45: delete “so called”

Line 49: change to “As with many…”

Line 51 change to “…in high energy environments…..(especially on their plateaus).”

Line 54: change to “V. flagellum is mostly found along…”

Line 59: change to “…have been patchily observed….”

Line 63: change to “…slow to zero recovery……. As with other forest-forming….”

Line 71: I do not believe that the MSFD is an action for monitoring sensitive habitats. Firstly it is a directive, secondly I do not think that it spells out specific sensitive habitats. I think that the Habitats Directive and corresponding national legislation would do this. This also refers to the use of the MSFD in Line 262.

Line 78: probably towed cameras should be “drift/drop/towed”. Add a small “s” after ROVs

Line 79: change to “corals, for example, sea….”

Line 80 – plural “polyps”

Line 81, need to mention that it is a sea whip in the first part of the introduction and use the singular here not the plural (colonies is the relevant plural in the sentence)

Line 82 change to “…very flexible and the often bend backward….”

Line 83 change to “…planes with respect to….this reason, size assessment…”

Line 89:  change to “…that colony width…”

Line 91 change to “…regardless of their…”

Materials and Methods

Line 97: I think that the vessel name is Ranger, not the Oceana Ranger – it belongs to Oceana, but the NGO is not the official name of the vessel.

Line 98 delete “ on the homonymic volcanic island”

Line 99: change to “…V. flagellum which was the focus of this study.”

Line 112: You do not explain at all how this unit was established in the frame of the ROV view. This is extremely important to the estimation of population density. The area that you consider may change with slope, view, use of zoom and specifically height of the camera above the seabed. How can this be established with just 2 laser spots?

Line 126: change to “…using the Fiji plugin for…”

Line 130: how did you know they were not facing forwards or back in a single image?

Line 132: Posidonia leaves: were they always in the correct plane and absolutely adjacent to the measured parts of the coral. If they are in the wrong plane, viewed at an angle or at a different distance from the measured part, or old and partially degraded, this will be inaccurate. To go forward with this it would have been necessary to do your own calibration measurements with Posidonia – see the decimal places later)

Results

Line 172: 40 is the total of the 16 units analysed – what was the approximate area of the population covered.

Line 174: please note in the text here that the +/- is standard error (I had to go looking for it in the methods)

Line 183-4. The abrasion reason belongs in the discussion.

Line 185: change to “… entangled, so it was….”

Line 196. I am not sure of the validity to 5 decimal places from the methodology you use.

Line 204: change to “The size-frequency…..”

Line 206: change to “..represented by large adults…”

Discussion

Line 213: change to “… has been sporadically observed in the Mediterranean Sea….”

Line 217: do you have a reference for “that tends to occupy the top”

Line 224: I am not sure that the use of the word ‘skeins’ is clear, perhaps ‘entanglements’ or ‘interactions’ is better, or just change to “…and the results seem to enhance…”

Line 229: change to “…represent a measure of intraspecific…” or “represent a form of intraspecific…”

Line 235: change to “…small colonies with respect to the…”

Line 240: change to “…populations tending to show…”

Line 247 change to “…In addition branch dichotomization is not as uncommon as previously thought [1]…”

Line 253 change to “…impacts, such as fisheries, and global warming…” (global warming is arguably an anthropogenic impact, but conflated with natural warming)

Conclusions

Line 257: change to “…found in the Aeolian…”

Line 258: change to “…as well as adding to the knowledge base on the geographic..”

Line 265: I think that the Habitats Directive is more relevant here for protecting sensitive and important habitats.

Supplementary Material: unfortunately, this link is not working, so this material could not be reviewed.

Acknowledgements: as noted above for the name, I believe the vessel is Oceana’s RV Ranger

Author Response

We thank you the reviewer for the accurate and useful revision. We provide here a point-by-point reply to all the comments and suggestions.

The manuscript describes the identification and analysis of a whip coral population off the island of Stromboli in the Mediterranean Sea. The data was collected by ROV video. The analysis included estimation of population density and structure. The analysis of structure is made from image analysis, by creation of a relationship for length-width on part of the population then applying the relationship through width measurements to create a size frequency analysis for the whole sampled population. The manuscript is relevant to the journal, but I have some issues with validity of how the method is applied, mostly related to lacking a solid validation. The point of the methodology is that it is non-invasive so the sensitive species is not harmed during measurement, but in the overall analysis the method uses measurements created from a subset of the images/population to apply to the whole but limited population observed. There is no mention of validation, on any physical specimens or another population – this point needs to be absolutely addressed.

Reply: We thank you the reviewer for identifying this issue that, now, has been highlighted in the discussion. Considering that we are studying deep populations, where humans cannot dive, and that are fragile and sensitive to sampling procedures, it is impossible to have physical specimens for validation without affecting the studied population. Studies on inaccessible areas (e.g. mesophotic and deep-sea) cannot be easily replicated or validated, and this is commonly accepted. This means that the method is not 100% robust, of course, but it represents a first, solid step to be implemented in the future.

There are two further points also that need to be addressed because you are presenting a measurement methodology: accuracy and resolution - and this runs into error and error propagation: How sure are you that the laser spots were always in the correct plane and perpendicular to the camera nest to the coral for 21 colonies. (it would be useful in at least one of the photos to show the laser spots highlighted if not easily seen.

The measurements are primarily based/calibrated on two parallel laser spots set at 10 cm distance. In section 3.1 calculated measurements go to 5 decimal places. However no calibration accuracy measurement is given (just standard error on the derived measurements) What is the potential error in a range of consecutive images for the laser spots.

Error uncertainty propagated by the different issues. Errors behind calibrating the scale from the lasers or against Posidonia leaves, errors with the measurements with on-screen software,

I do accept that the errors are reduced or not important when the measurements are binned into the different and wide size classes – but this needs to be stated.

Reply: laser pointers are commonly used in this type of studies and there is an extensive scientific literature based on the use of laser for scaling (and the validity of this method too). As clearly explained in the text, sequences not optimal for the laser and the coral position were discarded. In fact, 21 out of more than 700 colonies were those perfectly aligned with the lasers in order to have a model as much robust as possible.

We agree that error propagation is not considered but it is quite negligible considering that the results are organized in large size classes of 20 cm. However, we highlighted this point in the methods and in the discussion. The mean value has been reduced to the second decimal centimetre.

Your method the major point to the paper is hardly discussed and the targeted results – density and size distribution are not compared in detail to other populations, leaving the reader to have to look up the references to find this information.

Reply: Discussions have been improved. The few possible comparisons have been done, although this method allows to assess, for the first time, the population structure of this species thus, to the best of our knowledge, there are not similar results to compare.

Other edits/comments:

Simple Summary

Line 20: change to : “… Results of the survey indicated that the Aeolian coral population was in good condition and…”

Reply: Done.

Abstract

Line 26: syntax problem, the population structure/status of the coral does not result in monitoring actions.

Reply: We changed as “The population structure of the habitat-forming corals can inform about the status of the habitat, representing an essential aspect to monitor”.

Line 26: change to “Most Mediterranean corals….”

Reply: Done.

Line 29: change to “…flexibility impedes the taking of direct …”

Reply: Done.

Line 30: change to “…Viminella flagellum in the Aeolian…”

Reply: Done.

Line 34: change to “…the presence of large adults…”

Reply: Done.

Introduction:

Line 43: WORMS gives your given species name as “uncertain” but the accepted name is Ellisella flagellum (Johnson, 1863). I do not think that “possibly synonimised” is a correct way of referring to the species. Can you please clarify.

Reply: We clarified as follow: “The taxonomic position of this species is currently considered uncertain and possibly attributed to the genus Ellisella, although it is still to be clarified and the binomial V. flagellum will be used in this paper.”

Line 45: delete “so called”

Reply: Done.

Line 49: change to “As with many…”

Reply: Done.

Line 51 change to “…in high energy environments…..(especially on their plateaus).”

Reply: Done.

Line 54: change to “V. flagellum is mostly found along…”

Reply: Done.

Line 59: change to “…have been patchily observed….”

Reply: Done.

Line 63: change to “…slow to zero recovery……. As with other forest-forming….”

Reply: Done.

Line 71: I do not believe that the MSFD is an action for monitoring sensitive habitats. Firstly it is a directive, secondly I do not think that it spells out specific sensitive habitats. I think that the Habitats Directive and corresponding national legislation would do this. This also refers to the use of the MSFD in Line 262.

Reply: We do not agree. A series of monitoring actions have been carried out since 6-7 years within the MFSD that targets specific habitats (e.g. Posidonia meadows; rhodolith beds;  coralligenous) while do not consider coral forests. On the contrary, the Habitat Directive concerns the identification of target habitats to protect.

Line 78: probably towed cameras should be “drift/drop/towed”. Add a small “s” after ROVs

Reply: Done. We modified as “towed/drop cameras and Remotely Operated Vehicles (ROVs)”. We did not add “drift” as drift cameras are more for pelagic fauna rather than benthos.

Line 79: change to “corals, for example, sea….”

Reply: Done.

Line 80 – plural “polyps”

Reply: We do not agree. The plural of polyp leaf is polyp leaves.

Line 81, need to mention that it is a sea whip in the first part of the introduction and use the singular here not the plural (colonies is the relevant plural in the sentence)

Reply: We deleted “colonies” in order to make the sentence more clear and linear. We also specified in the introduction that it is a whip coral.

Line 82 change to “…very flexible and the often bend backward….”

Reply: Done. We changed “fold” with “bend”

Line 83 change to “…planes with respect to….this reason, size assessment…”

Reply: Done.

Line 89:  change to “…that colony width…”

Reply: Done.

Line 91 change to “…regardless of their…”

Reply: Done.

Materials and Methods

Line 97: I think that the vessel name is Ranger, not the Oceana Ranger – it belongs to Oceana, but the NGO is not the official name of the vessel.

Reply: We agree.

Line 98 delete “ on the homonymic volcanic island”

Reply: Done.

Line 99: change to “…V. flagellum which was the focus of this study.”

Reply: Done.

Line 112: You do not explain at all how this unit was established in the frame of the ROV view. This is extremely important to the estimation of population density. The area that you consider may change with slope, view, use of zoom and specifically height of the camera above the seabed. How can this be established with just 2 laser spots?

Reply: We explained that we used laser pointers as size measurement and reference. There is an extensive literature based on the use of the lasers, so we think it is unnecessary to explain it in detail. Regardless for the zoom, the laser pointers provide a scale of 10 cm that can be used to calibrate the image with ImageJ software. Every image is calibrated with its own scale. Sequences with bad visibility or with a non-ideal orientation of the camera for a correct measurement are of course discarded. The area of 2.5 m2 is used also because it is quite small to be affected by perspective problems.

Line 126: change to “…using the Fiji plugin for…”

Reply: Done.

Line 130: how did you know they were not facing forwards or back in a single image?

Reply: This aspect is clearly visible from videos.

Line 132: Posidonia leaves: were they always in the correct plane and absolutely adjacent to the measured parts of the coral. If they are in the wrong plane, viewed at an angle or at a different distance from the measured part, or old and partially degraded, this will be inaccurate. To go forward with this it would have been necessary to do your own calibration measurements with Posidonia – see the decimal places later)

Reply: we rephrased and implemented this part according to the suggestions.

Results

Line 172: 40 is the total of the 16 units analysed – what was the approximate area of the population covered.

Reply: 40 m2 is the area occupied by the population in our survey. We do not know if the population is present also in nearby areas and to what extent.

Line 174: please note in the text here that the +/- is standard error (I had to go looking for it in the methods)

Reply: it is already specified in the methods, so we think it is redundant to specify it also in the results.

Line 183-4. The abrasion reason belongs in the discussion.

Reply: We prefer to anticipate here in the results this aspect that is then developed in the discussion.

Line 185: change to “… entangled, so it was….”

Reply: Done.

Line 196. I am not sure of the validity to 5 decimal places from the methodology you use.

Reply: Benefiting from the accuracy of the software and the standardization of the method used for width measurements (FWHM explained in the text), we were able to achieve a good accuracy through the software. The 5-decimal value (µw) is not a measured width, but the mean value of our measurements. However, we agree that is more fair to stay at the second decimal, which is within our measurement error.

Line 204: change to “The size-frequency…..”

Reply: Done.

Line 206: change to “..represented by large adults…”

Reply: Done.

Discussion

Line 213: change to “… has been sporadically observed in the Mediterranean Sea….”

Reply: Done.

Line 217: do you have a reference for “that tends to occupy the top”

Reply: No, this consideration is based on our observations and previous experience in the basin.

Line 224: I am not sure that the use of the word ‘skeins’ is clear, perhaps ‘entanglements’ or ‘interactions’ is better, or just change to “…and the results seem to enhance…”

Reply: Done.

Line 229: change to “…represent a measure of intraspecific…” or “represent a form of intraspecific…”

Reply: Done.

Line 235: change to “…small colonies with respect to the…”

Reply: Done.

Line 240: change to “…populations tending to show…”

Reply: Done.

Line 247 change to “…In addition branch dichotomization is not as uncommon as previously thought [1]…”

Reply: Done.

Line 253 change to “…impacts, such as fisheries, and global warming…” (global warming is arguably an anthropogenic impact, but conflated with natural warming)

Reply: Done.

Conclusions

Line 257: change to “…found in the Aeolian…”

Reply: Done.

Line 258: change to “…as well as adding to the knowledge base on the geographic..”

Reply: for clarity, we prefer to keep the sentence as it is.

Line 265: I think that the Habitats Directive is more relevant here for protecting sensitive and important habitats.

Reply: We rephrased and mentioned both the directives, as follow: “Moreover, in the frame of current international initiatives targeting the monitoring of sensitive and important habitats and their conservation (e.g. EU Marine Strategy Framework Directive, EU Habitat Directive), these tools can inform and support appropriate actions for strategic environmental protection and a sustainable management of human activities”.

Supplementary Material: unfortunately, this link is not working, so this material could not be reviewed.

Reply: there are no supplementary materials. Apologize, we forgot to delete the formula from the template.

Acknowledgements: as noted above for the name, I believe the vessel is Oceana’s RV Ranger

Reply: Done.

Reviewer 3 Report

Overview:

A nice and simple paper examining a monospecific community of octocorals in the Mediterranean, with a new method of examining colony heights included. This paper fits well with the journal, and should be acceptable after some revisions.

Major concerns:

  1. The English is places is a bit odd. In general, the paper is well written and easy to follow, but there are some areas that need a bit of brushing up. I have tried to mention as many as I could in the minor comments below, but the paper would benefit from having a colleague proofread it once before resubmission, in my opinion.
  2. Terminology, particularly taxonomic. In many areas, the authors use words like “corals” without any context or definition. In different cases, and for different workers, such terms have different meanings. I suggest including proper taxonomic information at first mention of all different organisms. I have also included comments below in the minor comments for some of these instances, but please check the paper carefully regarding this issue.
  3. Line 112: Videos – how many? How long? Some numbers are needed here. Numbers of images analyses etc should also be given in the M&M.
  4. I might be missing something, but the images in Figure 3 show to me somehow that not the entire lengths of some colonies were measured using your new methodMany of the colored lines in C, for example, seem to stop well before the colonies do. If so, this would affect your results by creating more small colonies (as you observed). Perhaps I am missing something? Some more explanation in the Legend, at least, is needed.

Minor comments:

  1. Simple summary first sentence: Define corals or use a more exact term here. Elongated and unbranched along could also refer to some anemones or even zoantharians, and corals – perhaps you mean octocorals?
  2. Abstract first sentence: alcyonaceans – but give names here too; I assume you mean Octocorallia? Also black corals – order Antipatharia.
  3. Line 26: delete “of” before Mediterranean.
  4. Somewhere in the Abstract you should give the size of the area of this forest that was examined.
  5. Line 44: I would add a reference for this uncertain taxonomic status, as this is an unusual situation. WoRMS should suffice.
  6. Line 49: As with many other…
  7. Lines 52-53: “also thanks to” is a bit awkward.
  8. Line 55: Canary Islands
  9. Line 56: particularly below 350 m and down to approximately 1000 m
  10. Line 58: down to 500 m
  11. Line 79: Again, here, give relevant scientific names. Also line 81 for sea pens.
  12. Line 97: km.
  13. Line 98: “that was considered in this study”
  14. Line 133: delete “however”
  15. Line 144: this sentence should be “colony widths” and also needs some editing.
  16. Line 147: I cannot understand what you mean by the phrase starting with “also when they were positioned…” Can you clarify this?
  17. Line 170: As the title is in italics, the species name should NOT be in italics, and underlined instead.
  18. Line 172: What are these sampling units? Please define them.
  19. Lines 218-219: Do you have any other information or observations that would allow you to speculate on other forests in the area? What does the seafloor of this area look like?
  20. Line 239: What types of corals are you referring to here?
  21. Line 258: The section from “as well” should be split into a new sentence.

Author Response

We are grateful to the reviewer for the comments and suggestions received. We provide here a point-by-point reply.

Overview:

A nice and simple paper examining a monospecific community of octocorals in the Mediterranean, with a new method of examining colony heights included. This paper fits well with the journal, and should be acceptable after some revisions.

Reply: We thank you the reviewer for the positive feedback.

Major concerns:

The English is places is a bit odd. In general, the paper is well written and easy to follow, but there are some areas that need a bit of brushing up. I have tried to mention as many as I could in the minor comments below, but the paper would benefit from having a colleague proofread it once before resubmission, in my opinion.

Terminology, particularly taxonomic. In many areas, the authors use words like “corals” without any context or definition. In different cases, and for different workers, such terms have different meanings. I suggest including proper taxonomic information at first mention of all different organisms. I have also included comments below in the minor comments for some of these instances, but please check the paper carefully regarding this issue.

Reply: We thank the reviewer for the effort in checking the English. We improved it following the suggestions as well as thanks to the proof-reading by a native English lecturer.

Considering the term “coral”, this has been clarified since the title, where the Order and the Class are specified. It is also clarified in the beginning of the text. We double-checked for clairity.

Line 112: Videos – how many? How long? Some numbers are needed here. Numbers of images analyses etc should also be given in the M&M.

Reply: We implemented the methods with this information. Numbers of images analysed are provided in the results.

I might be missing something, but the images in Figure 3 show to me somehow that not the entire lengths of some colonies were measured using your new method. Many of the colored lines in C, for example, seem to stop well before the colonies do. If so, this would affect your results by creating more small colonies (as you observed). Perhaps I am missing something? Some more explanation in the Legend, at least, is needed.

Reply: We recognize that the colonies are very difficult to be distinguished from a single image with a wide view, as in this case. We are completely sure of the size of the colonies highlighted with colours, which are the results of the observation from different points of view. The purpose of the figure is to provide a visual idea of how complicate is the real situation, how few colonies can be measured with usual techniques and how many more have been measured with our method. Sometimes they seems to be considered only in part, but it is a distortion related to the countless colonies on the background. Based on the comment of the reviewer, we decided to change figure 3 with a less confusing sequence for our purpose.

Minor comments:

Simple summary first sentence: Define corals or use a more exact term here. Elongated and unbranched along could also refer to some anemones or even zoantharians, and corals – perhaps you mean octocorals?

Reply: Done. We made it clearer.

Abstract first sentence: alcyonaceans – but give names here too; I assume you mean Octocorallia? Also black corals – order Antipatharia.

Reply: We agree. In general, the abstract is simple and jargon-free, but in this case there is also a “simple summary” so we can be more technical in the abstract. We rephrased as “Anthozoa of the order Alcyonacea and Antipatharia”

Line 26: delete “of” before Mediterranean.

Reply: Done.

Somewhere in the Abstract you should give the size of the area of this forest that was examined.

Reply: We cannot be sure about the total extension of the forest, and we focused our survey only on part of it.

Line 44: I would add a reference for this uncertain taxonomic status, as this is an unusual situation. WoRMS should suffice.

Reply: Unfortunately, there are no references on WoRMS about this taxonomic issue because it is still matter of debate. We clarified it as follow “The taxonomic position of this species is currently considered uncertain and possibly attributed to the genus Ellisella, although it is still to be clarified and the binomial V. flagellum will be used in this paper.”

Line 49: As with many other…

Reply: Done.

Lines 52-53: “also thanks to” is a bit awkward.

Reply: We rephrased and avoided “also thanks”.

Line 55: Canary Islands

Reply: Correct.

Line 56: particularly below 350 m and down to approximately 1000 m

Reply: Done.

Line 58: down to 500 m

Reply: Done.

Line 79: Again, here, give relevant scientific names. Also line 81 for sea pens.

Reply: We prefer not to specify the scientific names in order to make the text more readable. Species names are easily provided in the title and the text of the works cited.

Line 97: km.

Reply: Done.

Line 98: “that was considered in this study”

Reply: Done.

Line 133: delete “however”

Reply: Done.

Line 144: this sentence should be “colony widths” and also needs some editing.

Reply: Done.

Line 147: I cannot understand what you mean by the phrase starting with “also when they were positioned…” Can you clarify this?

Reply: we clarified as follow “… also when they were not present entirely in the same plane of the size references (e.g. colonies positioned backward or forward).”

Line 170: As the title is in italics, the species name should NOT be in italics, and underlined instead.

Reply: We agree. Previous experience with the journal suggest that they prefer to avoid underlining species names. We will double check it with them for the final style check.

Line 172: What are these sampling units? Please define them.

Reply: sampling units are 2.5 m2 areas of the seabed as explained in the methods.

Lines 218-219: Do you have any other information or observations that would allow you to speculate on other forests in the area? What does the seafloor of this area look like?

Reply: We specified that we did not observed other colonies on comparable seabed areas. The seafloor of the area is quite complex but it present other zones with high rocky pinnacles over detritic bottom.

Line 239: What types of corals are you referring to here?

Reply: as reported in the literature cited, it refers to a series of scleractinian corals belonging to the genera Agaricia, Diploria, Eusmilia, Madracis, Meandrina, Montastraea, Porites and Siderastrea. We think it is not relevant to deepen to such detail in the text.

Line 258: The section from “as well” should be split into a new sentence.

Reply: Done.

Round 2

Reviewer 2 Report

Re-review of: Chimienti et al. A new forest of the whip coral Viminella flagellum (Anthozoa, Alcyonacea) in the Mediterranean Sea: a non-invasive method to assess its population structure.

Overall: The authors have replied to all my comments and taken my major issues into consideration, dealing with them satisfactorily and I believe that the manuscript could be accepted for publication with some further minor changes noted below. I would congratulate the authors on their work.

Line 19: change to “…V. flagellum in the Aeolian….: - sorry missed it the first time (and a few others below)

Line 54: change to plural “…plateaus)…”

Line 82: change to “…corals, for example sea fans…”

Line 92: change to “…colony width does not vary…” and then instead of consistently do you mean “considerably”, because it is consistent - the syntax is a little strange.

Line 137 can you change to “P. oceanica leaves have an almost-constant width of 1 cm which may slightly vary by about 1 mm according to the season and the geographic area [41].”

Line 151-2: change to “…colony widths do not vary…”

Line 233: change to “…mechanical abrasion…”

Line 249: change to “…tending to show a peak..”

Line 257: change to “…previously thought [1]…”

Line 260: change to “…populations represents a…”

Line 264: change to “…0.35 cm is….”

Line 268 either “include improving the method” or “include the improvement of the method”

Line 276: change to “Its finding represents a…”

Line 277 (old line 258 comment): I’m still not happy with the syntax of “still incomplete understating” it is “the still incomplete knowledge that understates the geographic and bathymetric distribution” or did you actually mean “the still-incomplete understanding of the geographic and bathymetric distribution” – all the problem comes from the word understating.

Line 288: still has the link for the supplementary material – as you noted it needs deleting.

Line 302 change to “The Captain and …”

Line 422: “warming” not worming

Previous Comments and your replies (old line numbers)
Line 80: my mistake.

Line 78: I’ll let that be, but point out that drift cameras are used in benthic work in many applications.

Line 112 comment on sample unit area: I have worked with video and lasers. I do find it difficult to understand how you can define a precise area of 2.5 m2 from two laser dots seen in a horizontal plane with an obliquely forward looking camera on a variable morphology seabed. I accept that you put an error on it of +/- 0.2, to take this into account, but then should this not be added in to your abundance estimates?

Line 174: you lose nothing by putting in (+/-s.e.), but the reader can lose time in looking back to see what it is if not remembered/read at that time.

Line 258 – see above

Author Response

We sincerely thank you the reviewer for the kind an clear comments, as well as for the detail of the review. Please find below a point-by-point response to the second round of revision.

We include a version of the manuscript with all previous changes accepted, while new changes are again in track change mode.

Thank you for your help and your work.

Line 19: change to “…V. flagellum in the Aeolian….: - sorry missed it the first time (and a few others below)

Reply: Done.

Line 54: change to plural “…plateaus)…”

Reply: We used plateaux as plural.

Line 82: change to “…corals, for example sea fans…”

Reply: Done.

Line 92: change to “…colony width does not vary…” and then instead of consistently do you mean “considerably”, because it is consistent - the syntax is a little strange.

Reply: Done. We agree.

Line 137 can you change to “P. oceanica leaves have an almost-constant width of 1 cm which may slightly vary by about 1 mm according to the season and the geographic area [41].”

Reply: Done.

Line 151-2: change to “…colony widths do not vary…”

Reply: Done.

Line 233: change to “…mechanical abrasion…”

Reply: Done.

Line 249: change to “…tending to show a peak..”

Reply: Done.

Line 257: change to “…previously thought [1]…”

Reply: Done.

Line 260: change to “…populations represents a…”

Reply: Done.

Line 264: change to “…0.35 cm is….”

Reply: Done.

Line 268 either “include improving the method” or “include the improvement of the method”

Reply: Done.

Line 276: change to “Its finding represents a…”

Reply: Done.

Line 277 (old line 258 comment): I’m still not happy with the syntax of “still incomplete understating” it is “the still incomplete knowledge that understates the geographic and bathymetric distribution” or did you actually mean “the still-incomplete understanding of the geographic and bathymetric distribution” – all the problem comes from the word understating.

Reply: We agree. We modified the text accordingly.

Line 288: still has the link for the supplementary material – as you noted it needs deleting.

Reply: Done.

Line 302 change to “The Captain and …”

Reply: Done.

Line 422: “warming” not worming

Reply: Done.

Previous Comments and your replies (old line numbers)
Line 80: my mistake.

Reply: No problem

Line 78: I’ll let that be, but point out that drift cameras are used in benthic work in many applications.

Reply: Done. We included the drift cameras in order to be as much inclusive as possible.

Line 112 comment on sample unit area: I have worked with video and lasers. I do find it difficult to understand how you can define a precise area of 2.5 m2 from two laser dots seen in a horizontal plane with an obliquely forward looking camera on a variable morphology seabed. I accept that you put an error on it of +/- 0.2, to take this into account, but then should this not be added in to your abundance estimates?

Reply: The ideal is to use three lasers with a triangular arrangement, but they are not always available on ROVs. By using two lasers it is possible to achieve a quite good level of accuracy for this kind of studies, particularly when the camera is calibrated appropriately and the seabed is not extremely heterogeneous (e.g. not different inclinations and topographies). Laser beams or other bi-dimensional size references (e.g. Posidonia leaves, a ruler etc.), together with a known framing for the camera in use, allow to estimate the surface observed. This is intrinsic in the correct use of ROV and video-analysis software, needing of course experience. The calibration/framing of the camera is essential in the same way of many other a-priori known features of the equipment including positioning, gps, lasers, sound velocity etc. A certain error occur anyway, as we highlight. Considering the resolution of our results, a 0.2 m error in surface was considered negligible, as it does not affect density estimates.

Line 174: you lose nothing by putting in (+/-s.e.), but the reader can lose time in looking back to see what it is if not remembered/read at that time.

Reply: We agree. Done.

Line 258 – see above

Reply: Fixed according to the suggestions.